# Evaluation of Cooking Quality, Nutritional and Texture Characteristics of Pasta Added with Oat Bran and Apple Flour

**DOI:** 10.3390/foods8080299

**Published:** 2019-07-30

**Authors:** Vicente Espinosa-Solis, Paul Baruk Zamudio-Flores, Juan Manuel Tirado-Gallegos, Salvador Ramírez-Mancinas, Guadalupe Isela Olivas-Orozco, Miguel Espino-Díaz, María Hernández-González, Verónica Graciela García-Cano, Olalla Sánchez-Ortíz, José Juan Buenrostro-Figueroa, Ramiro Baeza-Jiménez

**Affiliations:** 1Coordinación Académica Región Huasteca Sur de la UASLP, Universidad Autónoma de San Luís Potosí, km 5 carretera Tamazunchale-San Martín, Tamazunchale, San Luís Potosí C.P. 79960, Mexico; 2Coordinación de Tecnología de Alimentos de la Zona Templada, Centro de Investigación en Alimentación y Desarrollo, A.C., Avenida Río Conchos s/n, Parque Industrial, Cd. Cuauhtémoc, Chihuahua C.P. 31570, Mexico; 3Facultad de Zootecnia y Ecología, Universidad Autónoma de Chihuahua, Periférico Francisco R. Almada, km 1, Chihuahua C.P. 31453, Mexico; 4Departamento de Ciencia y Tecnología de Alimentos, División de Ciencia Animal, Universidad Autónoma Agraria Antonio Narro, Calzada Antonio Narro 1923, Buenavista, Saltillo, Coahuila C.P. 25315, Mexico; 5Tecnológico Nacional de México/Instituto Tecnológico de Ciudad Cuauhtémoc, Avenida Tecnológico S/N, Cd. Cuauhtémoc, Chihuahua C.P. 31500, Mexico; 6Centro de Investigación en Alimentación y Desarrollo, A.C., Unidad Delicias, Av. Cuarta Sur 3820, Fraccionamiento Vencederos del Desierto, Cd. Delicias, Chihuahua C.P. 33809, Mexico

**Keywords:** pasta, apple flour, oat bran, starch digestibility, texture profile analysis

## Abstract

This study reports the effect of the addition of oat bran and apple flour on the cooking quality, digestibility, antioxidant, nutritional and texture characteristics of a spaghetti-type pasta. Pasta samples were prepared by substituting 50% of durum wheat semolina with oat bran (OBP) or apple flour (AFP). AFP presented higher cooking loss and water absorption index than the control pasta prepared with 100% durum wheat semolina (WSP). The supplementation of pasta with oat bran increased the total dietary fiber content (16.43% *w/w*, dw), while apple flour decreased the protein content (11.16% *w/w*, dw). There was no significant difference in the resistant starch content among all pasta samples. The pasta samples made with 50/50 durum wheat semolina/oat bran and 50/50 durum wheat semolina/apple flour increased the antioxidant activity by ≈46% and ≈97%, respectively. The OBP and AFP samples had a similar texture to the control pasta. A 50% replacement of durum wheat semolina with oat bran in a pasta formulation decreased the caloric content and digestibility of its starch components. These attributes found in the oat bran pasta and apple flour pasta make them a healthy choice for the diet of people with specific nutritional needs.

## 1. Introduction

Pasta is an important source of carbohydrates, a food low in sodium, fat, and cholesterol, and is considered one of the most consumed food worldwide [1]. Traditionally, pasta is made from durum wheat semolina and water; these ingredients are combined to form a homogeneous mixture [2]. A good quality pasta must be “*al dente*”, a term that describes pasta cooked to be firm to the bite. In general, pasta must meet the consumer’s requirements, such as color retention, a smooth surface, firmness, and elasticity. Besides, it must tolerate moderate overcooking, have minimal nutritional losses after cooking, and offer pleasant taste [3]. During pasta cooking, the gelatinization of starch and denaturation of proteins cause the main structural changes in pasta’s texture [4,5]. There are different investigations regarding increasing the level of dietary fiber and reducing the glycemic index of pasta [6,7]. Dietary fiber has become of public interest due to its physiological and metabolic effects associated with a good state of health for humans [8]. Dietary fiber increases fecal volume, reduces intestinal transit time, and it can contribute to the antioxidant capacity of a product [9,10,11]. In this sense, there is research potential concerning the use of oat bran and apple flour. Oat bran is a high source of dietary fiber [12] and it can be used to provide functional properties to foods.

Oat bran has different types of antioxidants such as tocopherols and tocotrienols, sterols, avenanthramides, p-hydroxybenzoic acid, and vanillic acid [13]. On the other hand, apple fruits are rich in flavonoids, and their pulp contains essential amounts of hydroxycinnamic acid derivatives, mainly chlorogenic acid. Apple peel is considered a significant source of phenolic compounds, especially flavonols and anthocyanins [14]. Apple fruits have been used to obtain flours that preserve antioxidant capacity [15,16]. Phenolic compounds have a wide variety of biological properties such as antioxidant, antimicrobial, antifungal and anticarcinogenic properties [17,18,19,20]. Moreover, phenolic compounds have been related to a reduction in the aggregation of low-density lipoproteins (LDL) [21], and some epidemiological studies have linked apple consumption to a reduced risk for developing asthma and lung cancer [22].

To the best of our knowledge, there are no investigations performed in pasta added with a high content of oat bran or apple flour, ingredients that are high in fiber or bioactive compounds. In this regard, Bustos et al. [23] conducted a study to evaluate the effect of fiber addition on the sensory and nutritional characteristics of spaghetti-type pasta and concluded that a 10% replacement of wheat flour with oat bran did not increase the resistant starch content of pasta. Generally, many consumers do not like to eat enough cereals or raw fruits. The consumption of pasta supplemented with functional ingredients, such as oat bran or apple flour can play an essential role in achieving health benefits. This study aimed to evaluate the effect of supplementation of a spaghetti-type pasta with oat bran or apple flour on the cooking quality, nutritional and textural characteristics of durum wheat semolina pasta.

## 2. Materials and Methods

### 2.1. Raw Materials

The durum wheat semolina (Harinera San Blas, Puebla, Mexico), oat bran (El Granero Integral, Biogran, Spain) and unripe apples from the Golden Delicious cultivar were acquired in a commercial store in Cuauhtemoc City (Chihuahua, Mexico).

### 2.2. Apple Flour Preparation

The unripe apples were washed with potable water and cut into cubes of 1 × 1 × 1 cm, then immediately rinsed in a citric acid solution (0.3%, *w*/*v*) to avoid enzymatic darkening. The apple cubes were dried in a convection oven (1350 GM, VWR Scientific Inc., Cornelius, OR, USA) at 50 °C for 4 h, and grounded using a commercial mill (Mapisa Internacional S.A. de C.V., Mexico). The unripe apple flour was sieved on a 50 mesh screen (0.028 mm) and stored at 25 °C in sealed plastic containers. It is important to mention that whole apples were used to obtain the flour, only discarding the non-edible parts (seeds and peduncle).

### 2.3. Pasta Formulation and Processing

Three formulations were used for making spaghetti-type pasta (Table 1). The control pasta (WSP) was made using only durum wheat semolina, oat bran pasta (OBP) was prepared by replacing 50% of durum wheat semolina with oat bran, and the apple flour pasta (AFP) was made by replacing 50% of durum wheat semolina with apple flour. All formulations were mixed with 50 mL of water and left to stand for 12 min to allow the ingredients to hydrate. The spaghetti-type pasta extrudes (1.5 mm in diameter) were elaborated employing a commercial pasta machine (Kitchen Aid, Model KPRA, St. Joseph, MI, USA). The pasta was cut off the roller and dried at 45 °C for 4 h. Each pasta formulation was prepared in triplicate. The dry pasta samples were stored in sealed polyethylene bags at room temperature until they were analyzed.

### 2.4. Pasta Cooking Quality

The pasta cooking quality was determined by the AACC International Approved Method 66-50.01 [24]. Pasta samples were cooked in boiling distilled water. The optimal cooking time (OCT) was reached when the white center, in the cooked pasta’s center core, disappeared after compressing it between two pieces of glass at intervals of 30 s. The cooking loss (CL), expressed as grams of matter loss/100 g of raw pasta, was evaluated by determining the amount of solids lost into the cooking water and the water absorption index (WAI), expressed as the increment of pasta weight during cooking, was evaluated by weighing pasta before and after cooking. The WAI was determined using Equation (1).
(1)$$WAI=\;\left(\frac{\mathrm{Weigh}\;\mathrm{of}\;\mathrm{cooked}\;\mathrm{pasta}\;\left(g\right)\;-\;\mathrm{Weigh}\;\mathrm{of}\;\mathrm{uncooked}\;\mathrm{pasta}\;(g)}{\mathrm{Weigh}\;\mathrm{of}\;\mathrm{uncooked}\;\mathrm{pasta}\;(g)}\right)\times 100$$

### 2.5. Chemical Composition

For the determination of the chemical composition of pasta, samples were cooked at their optimal cooking time, subsequently frozen with liquid nitrogen and then dried using lyophilization. The dry samples were grounded and sieved before analysis. The ash, protein, and lipid contents were analyzed according to the AACC methods 08-01.01, 46-13.01, and 30-25.01 [24], respectively. The total dietary fiber content (TDF) was evaluated using the AOAC method 985.29 [25]. For comparison purposes, these determinations were also made on durum wheat semolina, oat bran, and unripe apple flour. These analyses were carried out in triplicate.

### 2.6. Antioxidant Analysis

The extraction of bioactive compounds from flours and cooked pasta were carried out using the method reported by Leyva-Corral et al. [15]. The total phenolic content (TPC) of the samples was determined following the modified colorimetric method of Folin–Ciocalteu [26]. The measurements were compared to a standard curve of gallic acid and expressed as mg of gallic acid equivalent/100 g of the dry sample (mg GAE/100 g dw). The scavenging capacity (SC) of the samples was determined using the ABTS [2,2′-azinobis-(3-ethylbenzothiazoline-6-sulfonate)] method [27]. A standard curve of trolox was elaborated and the inhibition percentage was calculated with Equation (2). The scavenging capacity of the samples was expressed as μmol of trolox equivalent/100 g of the dry sample (μmol TE/100 g dw).
(2)$$Inhibition\;\left(\%\right)=\;\left(1-\frac{absorbance\;after\;7\min reaction}{initial\;absorbance\;without\;sample}\right)\;\times \;100$$

### 2.7. Total Starch, Resistant Starch and In Vitro Digestibility of Cooked Pasta

The total starch (TS) and resistant starch (RS) contents were determined using the methodology reported by Goñi et al. [28]. For the determination of the TS, an enzymatic solution of amyloglucosidase (brand Roche, No. 102,857, Roche Diagnostics, Indianapolis, IN, USA) was used. The RS content was quantified using different enzyme solutions and incubation times as follows, 60 min incubation with pepsin, 16 h incubation with α-amylase, and 45 min incubation with amyloglucosidase. The glucose released by enzymatic digestion was quantified using the glucose oxidase/peroxidase assay (SERA-PAK^®^ Plus, Bayer de Mexico, SA de CV), reading the optical densities of the samples at 510 nm. The percentage of starch hydrolysis (the rate of in vitro digestion) was determined according to the methodology reported by Zamudio-Flores et al. [29].

### 2.8. Textural Characteristics

The hardness, cohesiveness, gumminess, elasticity, chewiness, and adhesiveness parameters of cooked pasta were evaluated by a texture profile analysis (TPA) with a TA-XT2i Texturometer (Stable Micro System, Godalming, UK) equipped with a 30 kg load cell. A 25 mm diameter cylindrical probe (made of aluminum, part code P/25, batch no. 12,500) was used to compress single pasta sheets at a constant deformation rate of 1 mm/s to 80% of the initial strand thickness. An automatic trigger force (10 g_f_ according to the load cell used) was set. All samples were prepared and maintained until the analysis according to the AACC approved method 66-50.01 [24]. At least four measurements were performed per treatment and the averages were reported.

### 2.9. Statistical Analysis

A completely randomized design was used to evaluate the effect of the addition of unripe apple flour and oat bran on the quality of wheat semolina pasta. For the analysis of the data, a one-way analysis of variance (ANOVA) was performed using SPSS software version 6.0 (SPSS Institute Inc., Cary, NC, USA). When significant differences were observed, a mean comparison was made by the Tukey multiple range procedure (*p* ≤ 0.05).

## 3. Results and Discussion

### 3.1. Cooking Quality of Pasta

The results obtained for the cooking quality of pasta added with oat bran and unripe apple flour are shown in Table 2. The control pasta made with 100% durum wheat semolina showed a lower cooking loss value in other studies. Manthey and Schorno [30] reported cooking loss values of 6.40–6.50 g/100 g of raw spaghetti pasta and Hernández-Nava et al. [31] reported a cooking loss value of 6.50 g/100 g of raw spaghetti pasta. However, Brennan and Tudorica [7] reported a lower cooking loss value (0.93 g/100 g of raw spaghetti pasta). WSP presented lower cooking loss than OBP and AFP, and no significant statistical differences were observed (*p* > 0.05) between OBP and AFP. A cooking loss of 12% or less was considered acceptable and indicative of good quality pasta [32]. OBP and AFP can be considered pasta of good cooking quality since they presented a cooking loss value of 5.52 g and 5.78 g/100 g of raw pasta, respectively.

Spaghetti-type pasta added with some functional ingredients generally presents greater cooking loss than the control [33]. It has been reported that the addition of gluten-free flours in the manufacture of spaghetti dilutes the gluten strength of semolina. Consequently, the spaghetti structure will experience disruption and weakening, which allows the leaching of many soluble solids from the pasta into the cooking water [33]. The control pasta absorbed a lower amount of water than OBP and AFP (Table 2). After cooking, WSP increased 2.5 times its weight, while OBP and AFP increased 2.7 and 3.1 times their weight, respectively. Diverse factors can explain these results; a higher protein network formed by gluten in WSP prevents the diffusion of water into the starch granules, limiting their swelling [34], the botanic origin of starch granules that during cooking absorb water, and the content of soluble fibers, which increases the water-binding capacity of a food product. This last factor could be the main reason for a higher amount of water absorbed by AFP. The water absorption percentage calculated in this study for WSP is similar to those reported by other authors in spaghetti-type pasta [34,35].

### 3.2. Proximate Composition of Functional Flours and Cooked Pasta

Oat bran presented the highest ash content (6.68%), followed by unripe apple flour (4.58%) and durum wheat semolina (0.87%) (Table 3). The ash content found in oat bran and unripe apple flour did not match the values obtained for the pasta sample made of these ingredients, this difference between flour and pasta samples could be since during the cooking process some components of pasta tend to leach out of the pasta structure. OBP presented lower ash content than AFP (*p* < 0.05), which means that a higher amount of ash leached out of OBP, even though both kinds of pasta presented similar cooking loss values. The ash content of cooked pasta ranged from 0.84 to 2.28. Similar ash contents (0.44–0.58%) were reported in spaghetti-type pasta prepared with different durum wheat flours and with the addition of amaranth and lupine flours [33].

The AFP presented the lowest protein content (11.16%) out of the three cooked pasta, which was related to the low protein content found in the unripe apple flour (3.35%) (Table 3). No significant statistical differences were observed (*p* > 0.05) between the protein content of the OBP and WSP. Osorio-Díaz et al. [36] reported ≈12% of protein content in a spaghetti-type pasta made with 100% durum wheat flour, which is similar to the protein content in WSP. There are reports of different types of flours used in the preparation of pasta, which increases or decreases the protein content; this behavior is due to a dilution effect or concentration of proteins, by adding ingredients that have a lower or higher content of these biomolecules [33,37]. The lipid content of the wheat semolina pasta did not change by the enrichment with oat bran or unripe apple flour (Table 3).

The results for total dietary fiber (TDF) in flours and cooked pasta samples presented similar behaviors. When comparing WSP with OBP, a significant statistical increment (*p* < 0.05) was observed from 10 to 16% (which represented an increase of 61%). This increment was due to the high TDF content of oat bran (40.32%); on the other hand, WSP and AFP did not show significant statistical differences (*p* > 0.05). Increasing the TDF levels in food products, such as pasta, is considered necessary due to the functional effects that fiber brings to the small intestine. For example, foods with high viscosity containing fiber can mainly cause a low post-prandial glycemic response due to the delay in glucose uptake [38,39].

### 3.3. Antioxidant Properties of Flours and Cooked Pasta Samples

The evaluations of the total phenolic content (TPC) and scavenging capacity (SC) of flours and cooked pasta samples enriched with oat bran or unripe apple flour are presented in Table 3. The unripe apple flour presented the highest TPC content (548.80 mg GAE/100 g dw) and SC (367.81 µmol TE/100 g dw) in comparison with durum wheat semolina and oat bran. The results for the TPC and SC obtained for the flours utilized in the pasta formulations presented the same pattern as the pasta made out of these ingredients. The cooked AFP had the highest content of TPC (315.63 mg GAE/100 g dw) and SC (252.41 µmol TE/100 g dw), compared with the cooked WSP and OBP. In this regard, the scientific literature reports that the Golden Delicious apple has a high content of bioactive compounds such as anthocyanins, phenolic acids, dihydrochalcones, flavanols and flavonols [14], which are directly related to the greater antioxidant power of AFP. OBP presented higher values of TPC (289.47 mg GAE/100 g) and SC (187.25 μmol TE/100 g) than the control pasta, which was attributed to the content of bioactive substances that the pasta samples made with 50/50 durum wheat semolina compounds present in oat bran [13]. These results indicate that semolina/oat bran and 50/50 durum wheat semolina/apple flour could be used as a functional food product, due to the significant antioxidant capacity preserved after the cooking process of pasta.

### 3.4. In Vitro Digestibility

The results for total starch (TS) and resistant starch (RS) contents of cooked pasta are presented in Table 3. WSP and AFP did not present significant statistical differences (*p* > 0.05) in the TS content; while the OBP sample presented the lowest amount of total starch (57.61 g/100 g dw). There were no significant statistical differences (*p* > 0.05) between the total starch content of durum wheat semolina and unripe apple flour with the respective pasta made with these ingredients (WSP and AFP). Unripe apple flour presented high content of total starch (72.15%), due to the physiological maturity of apples (green stage), which are rich in starch. Stevenson et al. [40] studied the structural and functional characteristics of the starches from different unripe apple varieties. They isolated starch from unripe apples and reported a 47.5% (dry weight) starch content for the Golden Delicious variety. This result is lower than the amount of starch found in our study for unripe apple flour (72.15%).

The low content of starch present in OBP (considered low in comparison with WSP and AFP) did not affect its cooking characteristics due to the high protein content presented in the oat bran (≈17%). A high level of protein in pasta products is essential, since pasta with oat bran is a product low in digestible carbohydrates, with the capacity to induce a low glycemic response [4]. The consumption of this product may be suitable for people with reduced caloric requirements (people with obesity and diabetes). The content of RS did not show significant statistical differences (*p* > 0.05) among the pasta analyzed. However, the RS content of the three pasta studied here was lower than pasta added with pea flour (3.78%) [41].

The WSP presented the highest hydrolysis percentage followed by AFP, and OBP presented the lowest value (Figure 1). After 15 min of reaction, WSP presented a hydrolysis value of 79%, while the OBP presented a lower hydrolysis value (64%); however, a slight increase in hydrolysis values was observed later and after 45 min the hydrolysis remained constant in all three samples. This behavior coincides with the contents of TS and RS determined in the pasta. The microstructure of pasta is related to its physical compaction and could be responsible for limited accessibility of digestive enzymes to starch and digestible carbohydrates. Low rates of in vitro hydrolysis of starch are frequently related to a moderate post-prandial glycemic response in vivo [15,42], which is considered an essential factor in the management of diets due to alteration of metabolic conditions, such as diabetes [43].

### 3.5. Texture Analysis of Cooked Pasta

The pasta added with unripe apple flour presented the highest hardness value (73.58 N), while the pasta added with oat bran presented the lowest value (59.94 N). The hardness of both samples did not show significant statistical differences (*p* > 0.05) compared to the control sample (Table 4). The cohesiveness and chewiness parameters of the three kinds of pasta did not present significant statistical differences (*p* > 0.05). The hardness, cohesiveness, and adhesiveness are measurements of different aspects of the intermolecular forces between the starch molecules and swelling of the starch granules in the pasta matrix [44]. No significant statistical differences were observed (*p* > 0.05) in terms of adhesiveness (a measure of the strength between the pasta and the contact surface).

Cohesiveness is an indicator of how the sample is kept compact through cooking, and no significant difference was observed (*p* > 0.05) in the cohesiveness among WSP, OBP, and AFP (Table 4). In general, the addition of oat bran or unripe apple flour to the formulation of wheat semolina pasta did not modify the properties of adhesiveness and cohesiveness. The texture is important in the determination of final acceptance by consumers, and this is one of the predominant criteria for evaluating the quality of pasta [45]. A decrease in the cohesiveness or an increase in the adhesiveness of the pasta suggests changes in the texture quality of the pasta added with unripe apple flour or oat bran, and it could be used to know the possible acceptability of the product by consumers. Various texture studies have been reported in spaghetti-type pasta added with different types of flour [31,36].

## 4. Conclusions

An increase greater than 50% in the total dietary fiber content in the oat bran pasta was obtained concerning the control pasta. Slight changes were observed in the texture characteristics of pasta added with oat bran and unripe apple flour when compared with the control sample. The pasta with oat bran presented the lowest content of total starch; while the pasta added with unripe apple flour was not different from the control sample. All samples showed similar contents of resistant starch. The addition of the fiber-rich ingredients reduced the hydrolysis of the starch component. Compared with the control pasta, the pasta added with oat bran or unripe apple flour increased their antioxidant capacity. The pasta added with oat bran can be an alternative for people with special nutritional requirements, such as diabetic or overweight people, due to the high content of dietary fiber and low content of total starch.

## Figures and Tables

**Figure 1 foods-08-00299-f001:**
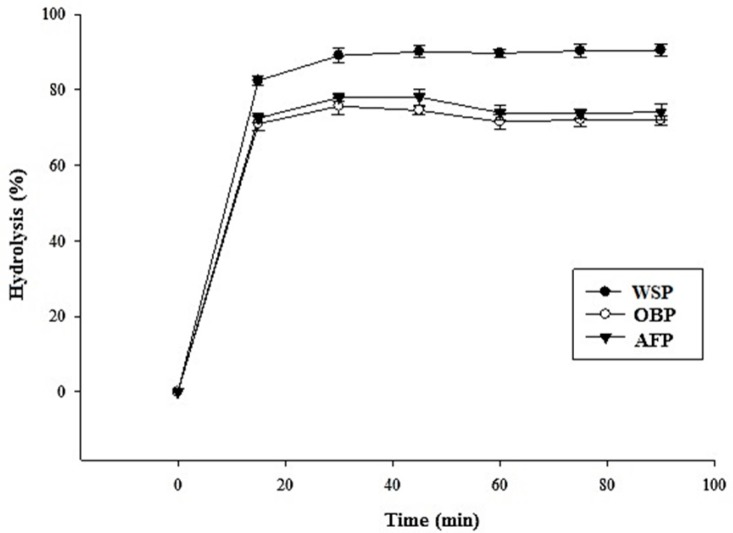
Rate of in vitro enzymatic hydrolysis of starch from the cooked pasta. Samples: WSP = control pasta made with 100% semolina (•); OBP = Pasta Sample made with 50/50 durum wheat semolina/oat bran (○) and AFP = Pasta sample made with 50/50 durum wheat semolina/apple flour (▼). The bars represent the standard error of the mean (*n* = 18).

**Table 1 foods-08-00299-t001:** Formulations of spaghetti-type pasta prepared with durum wheat semolina, oat bran and apple flour.

Ingredients (g/100g) ^1^	Pasta Type ^2^		
	WSP	OBP	AFP
Durum wheat semolina	100	50	50
Oat bran	0	50	0
Apple flour	0	0	50

^1^ Ingredients are expressed on a dry weight basis. Fifty mL of water were added per 100 g of mixture. ^2^ WSP = Control pasta made with 100% durum wheat semolina, OBP = Pasta Sample made with 50/50 durum wheat semolina/oat bran and AFP = Pasta sample made with 50/50 durum wheat semolina/apple flour.

**Table 2 foods-08-00299-t002:** Cooking quality parameters of wheat semolina pasta, oat bran enriched pasta and apple flour enriched pasta ^1^.

Pasta Type ^2^	Optimal Cooking Time (min)	Cooking Loss (g/100 g Raw Pasta)	Water Absorption Index (g/100 g Raw Pasta)
WSP	7.10 ± 0.30 ^b^	3.57 ± 0.26 ^b^	252.01 ± 2.49 ^c^
OBP	8.10 ± 0.50 ^a^	5.52 ± 0.14 ^a^	269.33 ± 3.99 ^b^
AFP	8.50 ± 0.20 ^a^	5.78 ± 0.17 ^a^	309.63 ± 2.56 ^a^

^1^ Results presented in the table are expressed as the mean value ± standard deviation (SD) for three replications. Different letters within the same column represent statistically significant differences (*p* < 0.05). ^2^ WSP = Control pasta made with 100% durum wheat semolina, OBP = Pasta Sample made with 50/50 durum wheat semolina/oat bran and AFP = Pasta sample made with 50/50 durum wheat semolina/apple flour.

**Table 3 foods-08-00299-t003:** Proximate composition and nutritional characteristics of flours and wheat semolina pasta, oat bran enriched pasta and apple flour enriched pasta ^1,2^.

Sample ^9^	Lipids (%)	Protein ^3^ (%)	Ash (%)	TDF ^4^ (%)	TS ^5^ (%)	RS ^6^ (%)	TPC ^7^ (mg GAE/100 g)	SC ^8^ (μmol TE/100 g)
WS	2.55 ± 0.26 ^b^	12.67 ± 0.21 ^b^	0.87 ± 0.02 ^e^	10.16 ± 0.37 ^c^	70.05 ± 0.43 ^a^	2.11 ± 0.11 ^b^	240.52 ± 5.56 ^f^	115.07 ± 6.36 ^f^
OB	3.82 ± 0.14 ^a^	16.94 ± 0.42 ^a^	6.68 ± 0.07 ^a^	40.32 ± 0.69 ^a^	13.04 ± 0.27 ^c^	1.88 ± 0.08 ^c^	476.20 ± 7.05 ^b^	305.81 ± 9.11 ^b^
AF	2.72 ± 0.09 ^b^	3.35 ± 0.40 ^d^	4.58 ± 0.02 ^b^	10.28 ± 0.28 ^c^	72.15 ± 0.27 ^a^	2.25 ± 0.11 ^b^	548.80 ± 5.80 ^a^	367.81 ± 8.81 ^a^
WSP *	2.67 ± 0.03 ^b^	12.78 ± 0.21 ^b^	0.84 ± 0.62 ^e^	10.20 ± 0.17 ^c^	71.07 ± 0.42 ^a^	2.75 ± 0.44 ^a^	255.25 ± 6.20 ^e^	128.32 ± 7.35 ^e^
OBP *	2.78 ± 0.16 ^b^	13.52 ± 0.62 ^b^	1.91 ± 0.13 ^d^	16.43 ± 0.22 ^b^	57.61 ± 0.31 ^b^	3.01 ± 0.15 ^a^	289.47 ± 8.85 ^d^	187.25 ± 5.83 ^d^
AFP *	2.61 ± 0.18 ^b^	11.16 ± 0.16 ^c^	2.28 ± 0.37 ^c^	10.68 ± 0.62 ^c^	71.81 ± 0.44 ^a^	2.89 ± 0.41 ^a^	315.63 ± 7.75 ^c^	252.41 ± 8.70 ^c^

^1^ Results presented in the table are expressed as the mean value ± standard deviation (SD) for three replications. Different letters within the same column represent statistically significant differences (*p* < 0.05). ^2^ Composition expressed on dry weight basis. ^3^ N × 5.85. ^4^ TDF = Total dietary fiber. ^5^ TS = Total starch. ^6^ RS=Resistant starch. ^7^ TPC = Total phenolic content. ^8^ SC = Scavenging capacity. ^9^ WS = Wheat semolina. OB = Oat bran. AF = Apple flour. WSP = Control pasta made with 100% durum wheat semolina. OBP = Pasta Sample made with 50/50 durum wheat semolina/oat bran and AFP = Pasta sample made with 50/50 durum wheat semolina/apple flour. *Pasta was cooked to the optimal cooking time and then lyophilized before analysis.

**Table 4 foods-08-00299-t004:** Texture profile analysis (TPA) of wheat semolina pasta (WSP), oat bran enriched pasta (OBP) and apple flour enriched pasta (AFP) ^1^.

Variable TPA	Pasta Type		
	WSP	OBP	AFP
Hardness (N)	67.98 ± 2.94 ^a,b^	59.94 ± 2.26 ^b^	73.58 ± 5.10 ^a^
Cohesiveness	0.49 ± 0.03 ^a^	0.48 ± 0.11 ^a^	0.45 ± 0.12 ^a^
Gumminess (N)	33.26 ± 0.20 ^a^	28.74 ± 0.10 ^b^	33.16 ± 0.29 ^a^
Elasticity	1.10 ± 0.05 ^b^	1.16 ± 0.03 ^a,b^	1.24 ± 0.06 ^a^
Chewiness (N)	0.04 ± 0.00 ^a^	0.03 ± 0.01 ^a^	0.04 ± 0.01 ^a^
Adhesiveness (N × m) 1 × 10^−9^	98.10 ± 0.00 ^b^	196.00 ± 98.10 ^a,b^	392.00 ± 98.10 ^a^

^1^ Results presented in the table are expressed as the mean value ± standard deviation (SD) for at least four replications. Different letters within the same row represent statistically significant differences (*P* < 0.05).

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
