# Peer review of "Evaluation of Cooking Quality, Nutritional and Texture Characteristics of Pasta Added with Oat Bran and Apple Flour"

_foods, 2019, doi:10.3390/foods8080299_

Round 1
Reviewer 1 Report
This manuscript seems to be sound and well prepared. I have some doubts about the extent of experiment. It limited, but acceptable.
More detailed comments you will find below
Keywords: replace golden deliccoius apple with apple flour.
Materials and Methods
2.2 Apple flour preparation
Did you peel apple before processing, did you discard other non edible parts?
2.3 Pasta formulation process
There is a little problem with table 1. For Durum wheat, Oat bran and Apple flour no amounts are given (g), but only share (%). In l. 98-99 a proportion water to flour is given, so it can be calculated, but in my opinion it should be amended.
2.4 Pasta cooking quality
WA or WAI ?
2.6 Antioxidant analysis
Eq. 2 is missing
2.8 Textural characteristics
Could you provide more details for this section, as AACC method directs to CC publication where Instron machine was applied. Could you provide a more detailed description about applied blade (catalogue number ?)
3.1 Cooking quality of pasta
168 reconsider ...in other studies
173 ref. 32 - nothing is mentioned about cooking loss
179 experiment or rather experience
184 the size and shape of starch granules - I would rather say starch origin (of course it influences the size and shape)
185 reconsider the phrase about SDF binding water
Table 4 I suggest to use N in place of g and kg. Also adhesivness I believe should be also in SI units. In fact It can be treated aswork (Physics)
Author Response
Response to Reviewer #1
Keywords: replace golden delicious apple with apple flour.
Response: The suggestion was accepted. Please see page 1, line 42 in green.
Materials and Methods
2.2 Apple flour preparation
Did you peel apple before processing, did you discard other non edible parts?
Response: To obtain the apple flour, the whole apples were used, including the peel. Only non-edible parts such as seeds and peduncle were discarded. Please see the clarification in page 3, lines 92-94 in green.
2.3 Pasta formulation process
There is a little problem with table 1. For Durum wheat, Oat bran and Apple flour no amounts are given (g), but only share (%). In l. 98-99 a proportion water to flour is given, so it can be calculated, but in my opinion it should be amended.
Response: Table 1 was modified to keep only the percentage of flour ingredients used in each formulation. Water content used for making pasta dough was removed from Table 1. Lines 98-99 were re-written. See page 3, line 100 and Table 1 in green.
2.4 Pasta cooking quality
WA or WAI ?
Response: We apologize for the mistakes, omissions and typos. The correct abbreviation is WAI = Water Absorption Index. See page 3, lines 117-118 and 119 in green.
2.6 Antioxidant analysis Eq. 2 is missing
Response: The equation have been added. See page 4, line 141 in green.
2.8 Textural characteristics
Could you provide more details for this section, as AACC method directs to CC publication where Instron machine was applied. Could you provide a more detailed description about applied blade (catalogue number ?)
Response: Additional information was added in this regard. See page 4, lines 157-160 and 161-162 in green.
3.1 Cooking quality of pasta
168 reconsider ...in other studies
179 experiment or rather experience
184 the size and shape of starch granules - I would rather say starch origin (of course it influences the size and shape)
185 reconsider the phrase about SDF binding water
Response: All the requested corrections have been made. See page 5, lines 176, 187, 192 and 193 in green.
173 ref. 32 - nothing is mentioned about cooking loss
Response: We apologize for the mistake, the reference Shogren et al. (2006) was replaced by the correct reference. See page 11, lines 438-439.
Table 4 I suggest to use N in place of g and kg. Also adhesivness I believe should be also in SI units. In fact It can be treated aswork (Physics)
Response: The suggestion was attended. See page 8 lines 309-310 and Table 4 in green.

Reviewer 2 Report
It was a bit difficult to read the manuscript as come of the sentences were a bit difficult to understand. The authors have tried their best to put the manuscript together but will suggest they look at some of the sentences again.
It will also adequate if the authors report on Rapidly and slowly digestible starch since determined the in vitro starch digestibility.
CORRECTIONS
Line 31: Rephrase “Pasta sample added with flour….” For clarity.
Line 32: Remove “was” from the sentence.
Line 37: Rephrase “Enrichment of pasta with 50% oat” for clarity. The statement does not imply a pasta sample with 50/50 durum wheat/oat.
Line 50: Remove “a” from the sentence.
Line 52: I believe you were referring to “denaturation” and not “denaturalization” in the sentence.
Line 66: “related to a with” should be rephrased to make the sentence clear.
Lines 69 and 70: The statement should be rephrased for clarity. Phrases like “few works” could be replaced.
Line 75: “benefits to health” could be replaced with “health benefits”.
Line 77: Remove the comma before “apple flour”. The “and” between “cooking quality” and “nutritional” should be replaced with a comma. Change “texture” to “textural”.
Line 105: For Table 1, it would be good to note that the percentages are weight/weight (w/w).
Line 115: “express” should be changed tom “expressed”.
Line 116: “the WA was….” could be a separate sentence for clarity. Also, does “WA” mean the same thing as “WAI” used in the equation? If so, that should be clarified.
Line 127: Add “s” to triplicate.
Line 130: “by” should be replaced with “using”.
Line 131: Insert “the” between “of” and “samples”. Replace “were” with “was”.
Line 134: Insert “the” between “of” and “samples”.
Line 141: Insert the reference/author name(s) after “reported by”.
Line 143: Add “s” to “follow”. The comma after “follow” could be changed to a colon for clarity.
Line 147: Italicize “in vitro”.
Line 173: “for this….” should start a new sentence for clarity.
Line 178: Remove the comma before “dilutes”. “as a consequence” should be replaced with “consequently”, and should start a new sentence for clarity.
Line 182: Replace the comma after “results” with a semicolon.
Line 184: Replace the semicolon with a comma. Rephrase “the size and…” for clarity.
Line 186: Replace “responsible” with “reason”.
Lines 194 and 195: Similar to line 37, rephrase “…enriched with 50% oat bran” and “…enriched with 50% apple flour” for clarity. The statements do not imply a pasta sample with 50/50 durum wheat/oat bran or 50/50 durum wheat/apple flour.
Lines 199 – 202: Rephrase the sentence for clarity.
Line 205: “similar ash…” should start a new sentence.
Lines 214 - 215: Similar to line 37, rephrase “…enriched with 50% oat bran” and “…enriched with 50% apple flour” for clarity. The statements do not imply a pasta sample with 50/50 durum wheat/oat bran or 50/50 durum wheat/apple flour.
Line 217: Remove the comma before “out of..”.
Line 220: 12% of protein content? Rephrase the sentence for clarity.
Line 226: Remove the comma before “presented..”.
Line 228: “this increment..” should begin a new sentence. “owed” could be replaced with “due” for simplicity.
Line 248: See the corrections for lines 214 – 215.
Line 259: “they isolated…” should be a new sentence for clarity.
Line 261: “this result….” Should be a new sentence for clarity.
Line 264: The comma can be removed.
Line 271: Since “followed by” was used, “presented the lowest value” can be removed.
Line 273: Rephrase “a 64% of hydrolysis” for clarity. The statement beginning with “However” is unclear. It should be rephrased.
Line 277: Italicize “in vitro”.
Line 278: Italicize “in vivo”.
Line 302: “both samples’…..” should be a new sentence.
GENERAL COMENTS:
· This is a simple, yet interesting and relevant research idea.
· The authors seemed to struggle a bit with sentence construction and grammar.
· The proximate composition and the texture analysis sections of the discussion should be discussed in depth if possible.
· The conclusion could be better. The authors, for the most part, repeated the results and discussion in the conclusion.
· I think the authors should be given the opportunity to review and incorporate these corrections and suggestions.
Author Response
Response to Reviewer #2
Line 31: Rephrase “Pasta sample added with flour….” For clarity.
Line 32: Remove “was” from the sentence.
Line 37: Rephrase “Enrichment of pasta with 50% oat” for clarity. The statement does not imply a pasta sample with 50/50 durum wheat/oat.
Line 50: Remove “a” from the sentence.
Line 52: I believe you were referring to “denaturation” and not “denaturalization” in the sentence.
Line 66: “related to a with” should be rephrased to make the sentence clear.
Lines 69 and 70: The statement should be rephrased for clarity. Phrases like “few works” could be replaced.
Line 75: “benefits to health” could be replaced with “health benefits”.
Line 77: Remove the comma before “apple flour”. The “and” between “cooking quality” and “nutritional” should be replaced with a comma. Change “texture” to “textural”.
Line 105: For Table 1, it would be good to note that the percentages are weight/weight (w/w).
Line 115: “express” should be changed tom “expressed”.
Line 116: “the WA was….” could be a separate sentence for clarity. Also, does “WA” mean the same thing as “WAI” used in the equation? If so, that should be clarified.
Line 127: Add “s” to triplicate.
Line 130: “by” should be replaced with “using”.
Line 131: Insert “the” between “of” and “samples”. Replace “were” with “was”.
Line 134: Insert “the” between “of” and “samples”.
Line 141: Insert the reference/author name(s) after “reported by”.
Line 143: Add “s” to “follow”. The comma after “follow” could be changed to a colon for clarity.
Line 147: Italicize “in vitro”.
Line 173: “for this….” should start a new sentence for clarity.
Line 178: Remove the comma before “dilutes”. “as a consequence” should be replaced with “consequently”, and should start a new sentence for clarity.
Line 182: Replace the comma after “results” with a semicolon.
Line 184: Replace the semicolon with a comma. Rephrase “the size and…” for clarity.
Line 186: Replace “responsible” with “reason”.
Lines 194 and 195: Similar to line 37, rephrase “…enriched with 50% oat bran” and “…enriched with 50% apple flour” for clarity. The statements do not imply a pasta sample with 50/50 durum wheat/oat bran or 50/50 durum wheat/apple flour.
Lines 199 – 202: Rephrase the sentence for clarity.
Line 205: “similar ash…” should start a new sentence.
Lines 214 - 215: Similar to line 37, rephrase “…enriched with 50% oat bran” and “…enriched with 50% apple flour” for clarity. The statements do not imply a pasta sample with 50/50 durum wheat/oat bran or 50/50 durum wheat/apple flour.
Line 217: Remove the comma before “out of..”.
Line 220: 12% of protein content? Rephrase the sentence for clarity.
Line 226: Remove the comma before “presented..”.
Line 228: “this increment..” should begin a new sentence. “owed” could be replaced with “due” for simplicity.
Line 248: See the corrections for lines 214 – 215.
Line 259: “they isolated…” should be a new sentence for clarity.
Line 261: “this result….” Should be a new sentence for clarity.
Line 264: The comma can be removed.
Line 271: Since “followed by” was used, “presented the lowest value” can be removed.
Line 273: Rephrase “a 64% of hydrolysis” for clarity. The statement beginning with “However” is unclear. It should be rephrased.
Line 277: Italicize “in vitro”.
Line 278: Italicize “in vivo”
Line 302: “both samples’…..” should be a new sentence..
Response: We apologize for the mistakes, omissions and typos. All the requested corrections have been made. See pages 1, 2, 3, 4, 5, 6, 7 and 8 in yellow.
GENERAL COMENTS:
· This is a simple, yet interesting and relevant research idea.
Response: We appreciate this comment.
· The authors seemed to struggle a bit with sentence construction and grammar.
Response: We appreciate your comments and suggestions. We improve based on these observations.
· The proximate composition and the texture analysis sections of the discussion should be discussed in depth if possible.
Response: We appreciate this observation; however, the co-authors and I believe that these sections have already been discussed in depth and compared with the scientific literature.
· The conclusion could be better. The authors, for the most part, repeated the results and discussion in the conclusion.
Response: The conclusions were improved.
· I think the authors should be given the opportunity to review and incorporate these corrections and suggestions.
Response: The corrections and suggestions were accepted.
It will also adequate if the authors report on Rapidly and slowly digestible starch since determined the in vitro starch digestibility.
Response: We agree with the reviewer´s point of view; however, these data an information have already been submitted to another journal, therefore, we do not consider appropriate to mention this results in this publication.
